# SPEL: Structured Prediction for Entity Linking

**Hassan S. Shavarani**
School of Computing Science
Simon Fraser University
BC, Canada
sshavara@sfu.ca

**Anoop Sarkar**
School of Computing Science
Simon Fraser University
BC, Canada
anoop@sfu.ca

## Abstract

Entity linking is a prominent thread of research focused on structured data creation by linking spans of text to an ontology or knowledge source. We revisit the use of structured prediction for entity linking which classifies each individual input token as an entity, and aggregates the token predictions. Our system, called SPEL (Structured prediction for Entity Linking) is a state-of-the-art entity linking system that uses some new ideas to apply structured prediction to the task of entity linking including: two refined fine-tuning steps; a context sensitive prediction aggregation strategy; reduction of the size of the model's output vocabulary, and; we address a common problem in entity-linking systems where there is a training vs. inference tokenization mismatch. Our experiments show that we can outperform the state-of-the-art on the commonly used AIDA benchmark dataset for entity linking to Wikipedia. Our method is also very compute efficient in terms of number of parameters and speed of inference.

 https://github.com/shavarani/SpEL

## 1 Introduction

Knowledge bases, such as Wikipedia and Yago (Pellissier Tanon et al., 2020), are valuable resources that facilitate structured information extraction from textual data. Entity Linking (Shen et al., 2014) involves identifying text spans (mentions) and disambiguating the concept or knowledge base entry to which the mention is linked.

Entity linking can be viewed as three interlinked tasks (Broscheit, 2019; Poerner et al., 2020; van Hulst et al., 2020):

(1) *Mention Detection* (Nadeau and Sekine, 2007) to scan the raw text and identify the potential spans that may contain entity links.

(2) *Candidate Generation* (e.g. Fang et al., 2020) to match each potential span with a number of potential entity records in the knowledge base.

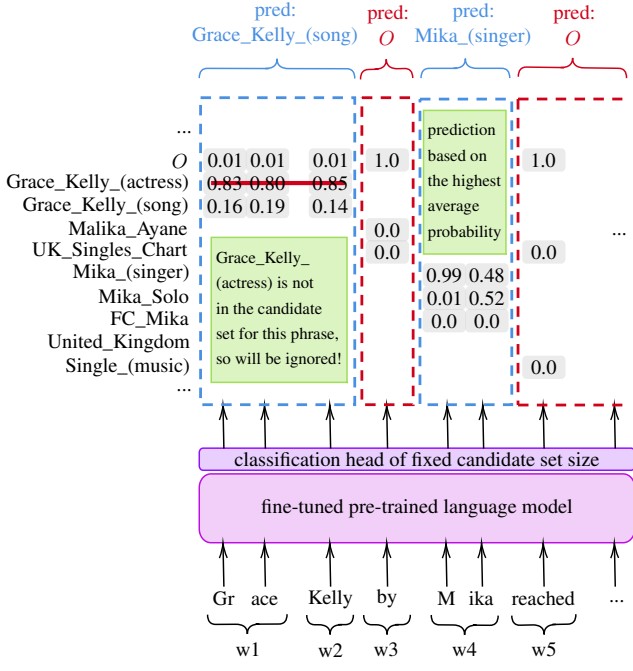

Figure 1: SPEL (Structured prediction for Entity Linking). In this example, we demonstrate top 3 most probable entities for each tokenized subword. In the first phrase, the most likely entity identified for `Grace Kelly` is incorrect. We use a candidate set for this mention to filter out irrelevant entity links with a high probability. In the second phrase, `Mika` is correctly narrowed down to the top 3 potential related entities. However, subword predictions do not consistently agree on the most probable prediction. In such cases, we calculate the average predicted probability for each entity across subwords and select the entity with the highest average probability.

(3) *Mention Disambiguation* (Ratinov et al., 2011; Yamada et al., 2022) to select one of the potential entity records for each detected mention.

An end-to-end entity linking system does all three tasks and links text spans to concepts. The system can either have independently modelled components (Piccinno and Ferragina, 2014; van Hulst et al., 2020) or jointly modelled components (Kolitsas et al., 2018; De Cao et al., 2021a).

Similar to almost all NLP tasks, recent en-

tity linking models use pre-trained representation learning methods that are based on Transformers (Vaswani et al., 2017). These methods commonly utilize bidirectional language models like BERT (Devlin et al., 2019) or auto-regressive causal language models such as GPT (Brown et al., 2020) or BART (Lewis et al., 2020), which are then fine-tuned on specific entity linking training datasets. In a number of such techniques, entity linking is framed as another well-studied problem, such as sequence-to-sequence translation (De Cao et al., 2021b) or question answering (Zhang et al., 2022).

Entity linking can be viewed as sequence tagging using structured prediction[1], aiming to assign one of the finitely many classes to every input utterance. This approach involves using a pre-trained model to encode each input subword token into a multi-layer context-aware dense vector representation. A classifier head calibrates each token representation to predict the entity for each subword token. Due to the large number of possible entities and issues with consistency of entity prediction across multiple subword tokens, structured prediction for entity linking (surprisingly) has not been studied in-depth. Our contributions in this paper are as follows:

(1) A new structured prediction framework for entity linking called SPEL (Figure 1). We demonstrate that SPEL establishes a new state-of-the-art for Wikipedia entity linking on the commonly used AIDA (Hoffart et al., 2011) dataset.

(2) Two separate and refined fine-tuning steps. One for general knowledge about Wikipedia concepts and one for specifically tuning on the AIDA dataset.

(3) A *context sensitive* prediction aggregation strategy that enables subword-level token classification while enforcing word-level and span-level prediction coherence. This does not incur additional inference time but drastically enhances the quality of the predicted annotation spans.

(4) We use the in-domain mention vocabulary to create a *fixed candidate set*. We use this to improve the efficiency and accuracy of entity linking.

---

[1] Structured prediction has shown successful enhancements in various NLP tasks, including Dependency Parsing (Zhou et al., 2015), Question Answering (Segal et al., 2020), and Machine Translation (Shavarani and Sarkar, 2021).

(5) Addressing the training/inference tokenization mismatch challenge in previous works which arises when differences in tokenization between training and testing phases lead to discrepancies. To address this challenge, we introduce an additional fine-tuning step where the model is fine-tuned on tokenized sequences without explicit mention location information. This encourages the model to learn robust representations that are not dependent on specific tokenization patterns, improving its generalization to the mention-agnostic tokenization at inference.

(6) New pre-training and inference ideas that can achieve a new state of the art with much better compute efficiency (fewer parameters) and much faster inference speed than previous methods.

(7) We have annotated and released AIDA/testc, a new entity linking test set for the AIDA dataset.

(8) Simplify and speed up the evaluation process of entity linking systems using the GERBIL platform (Röder et al., 2018) by providing a Python equivalent of its required Java middleware.

We introduce recent entity linking methods in Section 2; explain our approach to structured prediction for entity linking and our *context sensitive* prediction aggregation strategy in Section 3 with comprehensive experiments in Section 4.

## 2 Related Work

Entity linking can be framed as another well-studied task and the best solution for that task is applied. Autoregressive encoder-decoder sequence-to-sequence translation is one such approach. De Cao et al. (2021b) consider the input text as the source for translation and the text is annotated with Wikimedia markup containing the mention spans and the entity for each mention. Instead of mapping the entity identifiers into a single id (this is the default in many techniques including this work because the approach can be easily ported to ontologies other than Wikipedia), their model generates the entity label in a token-by-token basis (it generates the Wikipedia URL one subword at a time). The generation process follows a constrained decoding schema that prevents the model from producing invalid entity URLs while limiting

the generated output to the entities in a predefined candidate set (discussed in Section 3.1).

De Cao et al. (2021a) use a BERT-style bidirectional model fine-tuned to identify potential spans (mention detection) by learning spans using a *begin* probability and an *end* probability for each subword in the input text. For each potential span, they use a generative LSTM-based (Hochreiter and Schmidhuber, 1997) language model to generate the entity identifiers (token-by-token), and limit the generation process to pre-defined candidate sets.

Mrini et al. (2022) frame entity linking as a sequence-to-sequence translation task using BART (Lewis et al., 2020). They duplicate the BART decoder three times to fine-tune the model in a multitask setting. The two additional decoder modules are trained using auxiliary objectives of mention detection and re-ranking. While this training method increases the model size during training, they mitigate increased model size and speed at inference time by excluding the auxiliary decoder modules and employing sampling and re-ranking techniques on the generated target sequences.

Zhang et al. (2022) use Question Answering as a way to frame the entity linking task. They suggest a two-step entity linking model in which they use a fine-tuned Transformer-based BLINK (Wu et al., 2020) model to find all the potential entity records that might exist in the text and then utilize a fine-tuned question answering ELECTRA (Clark et al., 2020) model to identify the matching occurrences of the potential entities discovered in the first step. This approach obtains high accuracy; however, it is very resource intensive and inference is slow.

Structured prediction (subword token multi-label classification) is the other well-studied problem. Broscheit (2019) proposes a very simple entity linking model which places a classification head on top of a BERT language model and directly classifies each subword representation using a softmax over all the entities known to the model.

Our work also uses structured prediction because it is one of the most lightweight techniques in terms of compute cost and inference speed. We extend the structured prediction model in this work by (1) utilizing a *context sensitive* prediction aggregation strategy to form meaningful span annotations (Section 3.2), (2) addressing a training/inference tokenization mismatch issue (Section 3.3), (3) examining the role of different types of candidate sets (Section 3.1) in curating the predicted results,

and (4) optimizing the implementation of the structured prediction model. We obtain more than 6.1% points Micro-F1 improvement as well as more than 10 times reduction in required disk space, and close to 4 times reduction in the required GPU memory (in `base` case) compared to Broscheit (2019). Our approach is also faster at inference time than previous Transformer-based methods.

A number of recent techniques focus on enhancing the entity linking knowledge in BERT (or one of its variations) and utilize one or more such *knowledge-enhanced* models to perform the task of entity linking. Peters et al. (2019) inject Wikipedia and Wordnet information into the last few layers of BERT, Poerner et al. (2020) inject pre-trained Wikipedia2Vec (Yamada et al., 2020a) entity embeddings into the input layer of the language model while freezing the rest of its parameters, and Martins et al. (2019) leverage a Stack-LSTM (Dyer et al., 2015) Named Entity Recognition model to enhance entity linking performance using multitask learning to improve entity linking. These approaches are Wikipedia-centric and while we also experiment on Wikipedia exclusively, our approach can be used on any entity-linking dataset that has entities from other ontologies such as the MedMentions dataset (Mohan and Li, 2019) which links to concepts in UMLS ontology (Bodenreider, 2004).

Kolitsas et al. (2018) jointly model mention detection and mention disambiguation using an LSTM-based architecture while reusing the candidate sets created by Ganea and Hofmann (2017) as a replacement for the candidate generation step, and Kannan Ravi et al. (2021) follow a similar framework while modeling each of mention detection and mention disambiguation using separate BERT models. Feng et al. (2022) compute entity embeddings (instead of using pre-trained ones) using the average of the subword embeddings of the candidates and compare them to the average of the subword embeddings for the potential span (training a Siamese network; Bromley et al., 1993). Févry et al. (2020) investigate pre-training strategies specifically tailored for Transformer models to perform entity linking, diverging from the conventional use of pre-trained BERT models. And, van Hulst et al. (2020) propose a modular configuration that composes mention detection, candidate generation, and mention disambiguation in a pipeline approach, incorporating the most promising components from prior research.

In Section 4.4, we conduct experiments to compare SPEL to the methods discussed in this section.

## 3 Entity Linking

Formally, entity linking receives a passage (**p**) containing words $\{w_1, ..., w_n\}$ and produces a list containing $\ell$ span annotations. Each span annotation is expected to be a triplet of the form (*span start*, *span end*, *entity identifier*). The *span start* and *span end* values are expected to be character positions on the raw input **p**, and the *entity identifier* values are selected from a predefined vocabulary of entities (e.g. there would be approximately 6 million entities to choose from when entity linking to Wikipedia). The massive entity vocabulary size increases the model's hardware requirements and in some cases renders the task intractable.

### 3.1 Candidate Sets

To solve the entity vocabulary size problem, a common approach is to limit to $K$ most frequent entities in the knowledge base[2]. This vocabulary can be simply considered as the *fixed candidate set* for linking each mention to the knowledge base. Where no more information is available, the model will have to choose one entity from this *fixed candidate set*.

The selected *fixed candidate set* may lack many of the expected entity annotations at inference. Consequently, even if the model is highly capable, it may perform poorly during inference due to its inability to suggest the expected entities. Recognizing this challenge, there is a consensus among existing literature (Kolitsas et al., 2018; Broscheit, 2019; Peters et al., 2019; Poerner et al., 2020) to augment the *fixed candidate set* by including the expected entities necessary for inference. We adopt a similar approach to create the *fixed candidate set*, following the same line of reasoning as previous studies. Nonetheless, it's important to underscore that our adherence to this approach is driven by the desire for consistency with prior research; our framework, however, does not necessitate this specific method for effective functioning. In practical scenarios, one straightforward approach to construct the *fixed candidate set* is to base it on the anticipated entities (any subset of knowledge base entities that are pertinent to the task at hand) to be detected by SPEL.

An alternative is to use *mention-specific candidate sets* (Kolitsas et al., 2018; Peters et al., 2019; Kannan Ravi et al., 2021; De Cao et al., 2021b,a). *Mention-specific candidate sets* can be divided into two groups:

(1) *context-agnostic mention-specific sets* which are usually generated over large amounts of annotated text and try to model the probability of each mention span to all possible entity identifiers without assuming a specific context in which the mention would appear. KB+Yago[3] (Ganea and Hofmann, 2017) contains candidate lists for approximately 200K mentions created over the entire English Wikipedia combined with the Yago dictionary of (Hoffart et al., 2011).

(2) *context-aware mention-specific sets* can be constructed if there is a method for identifying mentions and a set of candidates for those mentions. For example, Pershina et al. (2015) have built such candidate sets, called PPRforNED[4]. Such lists have been primarily suggested for the task of entity disambiguation where the mention is provided in the input. As gold mentions are not available for real-world use cases of entity linking, this type of candidate sets have fallen out of favor.

*Mention-specific* candidate sets consist of many entity identifiers and the correct entity identifier is not guaranteed to exist given the mention span.

### 3.2 Structured Prediction for Entity Linking

For a sequence of subwords $S = \{s_1, s_2, ..., s_n\}$[5], we employ RoBERTa (Liu et al., 2019), in both `base` and `large` sizes, as our underlying model $M$ to encode $S$ into $H \in \mathbb{R}^{n \times d}$ where $d$ is the hidden representation dimension of $M$. Each representation $h_i \in H, i \in 1, ..., n$ is then transformed into a distribution over the *fixed candidate set* of size *KB* using a transformation matrix $W \in \mathbb{R}^{d \times KB}$. This results in $P_i = h_i W$, where $P_i$ represents the probability distribution for the $i^{th}$ subword in the input sequence.

When we set up fine-tuning for this task, we use hard negative mining (Gillick et al., 2019) to find the most probable incorrect predictions in the

---

[2]For Wikipedia, we can define an entity frequency as the number of times a title is hyperlinked in the other pages.

[3]https://github.com/yifding/deep_ed_PyTorch

[4]https://github.com/masha-p/PPRforNED

[5]When feeding a long text in training and inference, we split the text into smaller overlapping chunks.

batch[6]. In each fine-tuning step, we update the network based on the subword classification probabilities of the hard negative examples as well as the expected prediction. To increase inference speed, the classification head does not normalize the predicted scores.

We employ binary cross-entropy with logits, Equation (1), as our loss function, which is calculated over many factors. Let $N$ represent the total number of selected examples ($\psi$) comprising the one positive example corresponding to the expected prediction and the negative examples. Additionally, $a_{i,j}$ takes a value of 1 when the $j^{th}$ member of $\psi$ correctly points to the entity identifier for subword $s_i$, $p_{i,j}$ denotes model's predicted score for linking the $j^{th}$ member of the selected examples to the $i^{th}$ subword, and $\sigma$ is the sigmoid function, which maps the scores to $[0, 1]$.

$$\mathcal{L}_i = -\frac{1}{N} \sum_{j=1}^{N} \left[ a_{i,j} \cdot \log \left( \sigma(p_{i,j}) \right) \right.$$
$$\left. + (1 - a_{i,j}) \cdot \log \left( 1 - \sigma(p_{i,j}) \right) \right] \quad (1)$$

During inference we collect the top $k$ predictions for each subword $i$ based on the predicted probabilities in $P_i$. We then collect subwords that belong to the same word into a single group, which we call the *word annotation*. For each word annotation, we generate an aggregated entity identifier prediction set by taking the union of the entity identifiers predicted for the subwords. We then compute the weighted average of the prediction probabilities for each entity identifier to obtain the word-level probability score over entities.

Consecutive word-level entity labels when they refer to the same concept are joined into a single mention span over that phrase.

When a *mention-specific* candidate set is available, and the mention surface form matches one of the mentions in the candidate set, we filter out any predictions from the phrase annotation that are not present in the candidate set, regardless of their probability[7]. The final prediction for an entity span is generated based on the most probable prediction

in the phrase annotations, excluding the ones annotated with O (which means the phrase is not an entity). As an additional post-processing cleanup step, we reject phrase annotations that span over a single punctuation subword (e.g. a single period or comma) or a single function (sub)word (e.g. and, by, ...). In such cases, we manually override the model's prediction to O.

This *context sensitive* prediction aggregation strategy leads to improved performance and enhances prediction results in inference. Our strategy ensures that annotation spans do not begin or end inside a word[8], and the conflicts between the subword predictions within a word are resolved by the average prediction probability for each entity identifier.

This simpler method to ensure label consistency does better than using a CRF layer (Lafferty et al., 2001). Although our experiments show that a CRF layer does not improve our model, our readers can think of the suggested strategy as a domain-tailored, non-parametric, and rule-driven version of a CRF layer which guides the model to unify the predicted subword-level entity predictions considering the local context. Based on our experiments (Table 2), although we do not explicitly model Mention Detection (as predicting the *span start* and *span end* probability scores or separate BIO tags) for each subword in inference time, we observe a high in-domain accuracy in distinguishing O spans from non-O spans in predictions as a result of using the *context sensitive* prediction aggregation strategy.

Our modelling framework, SPEL, stands for *Structured Prediction for Entity Linking*.

### 3.3 Fine-tuning Procedure

Heinzerling and Inui (2021) argue that pre-trained language models can produce better representations when they are first fine-tuned on a much larger entity-linked training data (almost like a further pre-training step) and then subsequently fine-tuned for the entity-linking task. We perform such a multi-step fine-tuning procedure: first fine-tuning on a large dataset encompassing general knowledge on the set of linked concepts and then fine-tuning on an in-domain dataset specific to the target domain over which we aim to perform entity linking.

---

[6]We add random negative examples in addition to hard negatives to make sure we get to 5K negative examples for each batch when fine-tuning on CoNLL/AIDA and 10K negatives for general fine-tuning.

[7]The presence of a *mention-specific* candidate set is *not* a prerequisite for this technique to be effective.

[8]For instance, in the word U.S., if in the U.S part, the subwords have high likelihood for the concept The United States and the ending . refers to an O, the conflict is resolved so that the entire word U.S. is linked to The United States.

**General knowledge fine-tuning.** In the first step, we fine-tune the pre-trained language model using text that includes links to the knowledge base (in our experiments, we use a large subset of English Wikipedia[9]). As mentioned in (Broscheit, 2019), it helps if this data is aware of the mentions (using special space character subwords before and after each span that is linked to an entity). This helps the model learn the identification of the starting and ending subwords in entity mention spans. However, this imposes a mismatch in the distributions of the data in fine-tuning compared to inference, where the model does not have access to the entity mentions to perform the customized tokenization. To address this issue, as a subsequent fine-tuning step, we iterate again through the large entity-linked dataset which is re-tokenized *without* the knowledge of the mention spans.

**Domain specific fine-tuning.** In the third and last fine-tuning step, we refocus the model's attention to the in-domain dataset annotated with a *fixed candidate set* which usually is a subset of all the knowledge base entities that the model has observed in the previous two fine-tuning steps. Similar to the second fine-tuning step, we tokenize the in-domain dataset *without* the knowledge of the mention spans.

## 4 Experiments

### 4.1 Data

For our experiments, we focus on Wikipedia as the knowledge base and we use the following datasets for the fine-tuning steps mentioned in Section 3.3.

**Wikipedia** we use the 20230820 dump of Wikipedia (with approximately 238K documents), and we use the script from (Broscheit, 2019) to handle incomplete annotations, perform mention-aware customized tokenization, and compute the average probability of linking to no entity (called the *Nil* probability) for the 1000 most frequent entities. The *Nil* probability is used to modify the Wikipedia training data annotations in such a way that the chance of linking a surface form referring to a frequent entity to O is almost 0. We construct the Wikipedia *fixed candidate set* using the union of the 500K most frequent mentions in the Wikipedia dump and the *fixed candidate set* of AIDA and the test datasets. We split the content of Wikipedia pages into chunks consisting of 254 subwords with

a 20 subword overlap between consecutive chunks. After the split, our dataset contains 3,055,221 training instances with 1000 instances for validation. We also create a mention-agnostic re-tokenized version of this dataset with the same exact mentions to perform the second step of general knowledge fine-tuning as explained above.

**AIDA** (Hoffart et al., 2011) contains manual Wikipedia annotations for the 1393 Reuters news stories originally published for the CoNLL-2003 Named Entity Recognition Shared Task (Tjong Kim Sang and De Meulder, 2003). Its `train`, `testa`, and `testb` splits contain 946, 216, and 231 documents, respectively. It has a *fixed candidate set* size of 5600 (including O tag) and for evaluation on the AIDA test sets, we shrink the classification head in the model to these 5600 candidates and disregard the rest of the 500K candidates[10].

### 4.2 Evaluation using GERBIL

The GERBIL platform (Röder et al., 2018) is an evaluation toolkit (publicly available online) that eliminates any mistakes and allows for a fair comparison between methods. However, GERBIL is a Java toolkit, while most of modern entity linking work is done in Python. GERBIL developers recommend using `SpotWrapNifWS4Test` (a middleware tool written in Java) to connect Python entity linkers to GERBIL. Because of the complexity of this setup, we have not been able to directly evaluate some of the earlier publications due to software version mismatches and communication errors between Python and Java. This is a drawback that discourages researchers from using GERBIL. To address this issue, we provide a Python equivalent of `SpotWrapNifWS4Test` to encourage entity linking researchers to use GERBIL for fair repeatable comparisons. We evaluate all SPEL models using GERBIL in the `A2KB` experiment type, and report InKB strong annotation matching scores for entity linking. Only five of the publications to which we compare use GERBIL, however, all report InKB strong Micro-F1 scores allowing a direct comparison to our work.

### 4.3 Experimental Setup

For the first general knowledge fine-tuning step (Section 3.3), as a warm-up to full fine-tuning, we

---

[9]Limited to the articles that contain some presence of the entities in our selected *fixed candidate set*.

[10]Another implementation idea can revolve around multiplying the predicted output vector into a mask vector that masks all the candidates not in the expected 5600 entities.

freeze the entire ROBERTA model and only modify the classification head parameters on top of the encoder. We fine-tune with this configuration for 3 epochs and subsequently continue with fine-tuning all model parameters. We stop the fine-tuning process in this phase when the subword-level entity linking F1 score on the validation set shows no improvement for 2 consecutive epochs. Following this, we proceed to the second phase of full fine-tuning, where we fine-tune all model parameters using the mention-agnostic re-tokenized Wikipedia fine-tuning data. Just like phase one, we stop this phase based on the same criteria. We implement SPEL using `pytorch`, utilize `Adam` optimizer with a learning rate of $5e^{-5}$ to fine-tune the encoder parameters, and use `SparseAdam` optimizer with a learning rate of $0.01$ to fine-tune the classification head. We run fine-tuning phases one and two on the large subset of Wikipedia using two Nvidia Titan RTX GPUs.

For the last phase of fine-tuning on the AIDA dataset (Section 3.3), we freeze the first four layers of the encoder (including the embedding layer) as well as the shrunk classification head parameters, and we fine-tune the rest of the model parameters for 30 epochs (over the `train` part of AIDA dataset). We run this step using one Nvidia 1060 with 6 GBs of GPU memory, and accumulate gradients (Ott et al., 2018) for 4 batches before updating model parameters. We perform redirect normalization[11] on the final predicted spans.

### 4.4 Experiments and Results

In this section, we conduct experiments to evaluate the performance of both SPEL-base and SPEL-large (referring to the size of the underlying ROBERTA model) in different configurations concerning the use of candidate sets (Section 3.1), and report our experimental results over the AIDA test datasets in Table 1.

In the first configuration, we examine our model without any *mention-specific* candidate sets. Our results show a minimum of 5.3 Micro-F1 score improvement in AIDA test sets compared to (Broscheit, 2019) while significantly reducing the required parameter size on GPU by fourfold, resulting in a 7.2 times increase in inference speed in `base` case.

Next, we run SPEL in three other configu-

rations: (1) utilizing the KB+Yago (Ganea and Hofmann, 2017) context-agnostic candidate set, (2) employing the PPRforNED (Pershina et al., 2015) context-aware candidate set, and (3) adapting PPRforNED to aggregate the candidate information for each mention surface form, resulting in a context-agnostic candidate set, excluding context-specific information.

Candidate sets help reject many over-generated spans. If a mention's candidate set is empty, the mention span is excluded from further consideration. While this approach typically leads to improved precision and subsequent enhancement in F1 score, instances may arise where the model correctly predicts mentions that are not encompassed within the candidate sets. This can lead to lower recall in the evaluation. The observed Micro-F1 score drop when employing KB+Yago candidate sets compared to the scenario where no *mention-specific* candidate set is utilized, can be attributed to these cases.

SPEL-large using context-aware candidate sets achieves the highest boost, reporting 2.1 and 2.3 Micro-F1 scores improvement over `testa` and `testb` sets of AIDA, respectively, and establishes a new state-of-the-art for AIDA dataset. It is noteworthy to consider that the proposed model by Zhang et al. (2022) demands significant computational resources, including tens of gigabytes of RAM and over 7 and 2.7 times the number of parameters on GPU compared to SPEL-base and SPEL-large, respectively. Despite these resource-intensive requirements, SPEL outperforms Zhang et al. (2022). The comparison between our results and that of De Cao et al. (2021a,b) demonstrates that generating entity descriptions (which can share lexical information with the mention text) is not necessary even for high accuracy Wikipedia entity linking. Our approach can be easily extended to ontologies without textual concept descriptions, while methods that generate entity descriptions cannot.

Lastly, in Table 2, we compare SPEL-base, which utilizes the *context sensitive* prediction aggregation strategy to convert subword-level predicted entity identifiers into span-level predictions, to the model proposed by De Cao et al. (2021a). The latter model explicitly models the start and end positions of the spans for mention detection. We employ the evaluation script released by De Cao et al. (2021a) for our assessment. The results confirm that, despite not using `BIO` tags or ex-

---

[11]See Appendix A for more on this standard normalization technique.

| Approach | EL Micro-F1 | | #params on GPU | speed sec/doc |
|---|---|---|---|---|
| | testa | testb | | |
| Hoffart et al. (2011) (Linear) | 72.4 | 72.8 | - | - |
| Kolitsas et al. (2018) (LSTM) | 89.4 | 82.4 | 330.7M | 0.097 |
| Broscheit (2019) (BERT) | 86.0 | 79.3 | 495.1M | 0.613 |
| Peters et al. (2019) (BERT) | 82.1 | 73.7 | - | - |
| Martins et al. (2019) (Stack-LSTM) | 85.2 | 81.9 | - | - |
| van Hulst et al. (2020) (LSTM) | 83.3 | 82.4 | 19.0M | 0.337 |
| Févry et al. (2020) (Transformer) | 79.7 | 76.7 | - | - |
| Poerner et al. (2020) (BERT) | 90.8 | 85.0 | 131.1M | - |
| Kannan Ravi et al. (2021) (BERT) | - | 83.1 | - | - |
| De Cao et al. (2021b) (BART) | - | 83.7 | 406.3M | 40.969 |
| De Cao et al. (2021a) (RoBERTa+LSTM) | | | | |
| (no mention-specific candidate set) | 61.9 | 49.4 | 124.8M | 0.268 |
| (using PPRforNED candidate set) | 90.1 | 85.5 | 124.8M | 0.194 |
| Mrini et al. (2022) (BART) | - | 85.7 | (train) 811.5M (test) 406.2M | - |
| Zhang et al. (2022) (BLINK+ELECTRA) | - | 85.8 | 1004.3M | - |
| Feng et al. (2022) (BERT) | - | 86.3 | 157.3M | - |
| SPEL-base (no mention-specific candidate set) | 91.3 | 85.5 | 128.9M | 0.084 |
| SPEL-base (KB+Yago candidate set) | 90.6 | 85.7 | 128.9M | 0.158 |
| SPEL-base (PPRforNED candidate set) | | | | |
| context-agnostic | 91.7 | 86.8 | 128.9M | 0.156 |
| context-aware | 92.7 | 88.1 | 128.9M | 0.156 |
| SPEL-large (no mention-specific candidate set) | 91.6 | 85.8 | 361.1M | 0.273 |
| SPEL-large (KB+Yago candidate set) | 90.8 | 85.7 | 361.1M | 0.267 |
| SPEL-large (PPRforNED candidate set) | | | | |
| context-agnostic | 92.0 | 87.3 | 361.1M | 0.268 |
| context-aware | **92.9** | **88.6** | 361.1M | 0.267 |

Table 1: Entity Linking evaluation results of SPEL compared to that of the literature over AIDA test sets. *#params on GPU* only considers the total number of parameters that will directly effect the cost of GPU acquisition and does not reflect upon the total amount of data loaded into/from main memory and disk.

plicitly modeling span boundaries, SPEL demonstrates strong performance in mention detection, with a high level of accuracy. Its near-perfect precision scores indicate its ability to minimize overgenerated predictions, contributing to its state-of-the-art entity linking performance.

### 4.5 Comparison to OpenAI GPT

Large generative language models are effective zero shot and few shot learners (Brown et al., 2020) at many NLP tasks. We evaluate GPT-3.5 and GPT-4 for the task of entity linking using various prompts. We frame the problem for the generative LM as in De Cao et al. (2021b), where it produces markup around the mentions (described in detail in Appendix B). Table 3 compares the GPT evaluation results to that of SPEL. For a fair comparison, we consider the evaluation results without any *mention-specific* candidate sets. Currently the results are much worse than the state-of-the-art and using GPT-4 is more expensive. Further research

into few-shot in-context learning on GPT-4 is likely to improve these results since LLMs have extensive knowledge about entities but cannot directly reason about specific Wikipedia URLs[12].

### 4.6 Practicality of the Fixed Candidate Sets

A valid concern regarding SPEL pertains to the construction of the *fixed candidate set* and its practicality in real-world scenarios, where the testing data may not be predetermined, making it challenging when attempting to assemble a subset of knowledge base entries for this purpose. As mentioned in Section 3.1, it is possible to construct this set based on the expected entities that SPEL should detect. In this section, we take a more flexible approach, and consider the entire set of 500K general fine-tuning entities as the *fixed candidate set*.

Furthermore, taking inspiration from Liu and Ritter (2023) regarding the extended existence of the CoNLL-2003 dataset, and consequently the AIDA

---
[12]See (Cho et al., 2022) for some ways to address this issue.

| Approach | MD Micro Scores | | | | | |
| --- | --- | --- | --- | --- | --- | --- |
| | testa | | | testb | | |
| | P | R | F1 | P | R | F1 |
| De Cao et al. (2021a) (using PPRforNED c. set) | 93.9 | 96.7 | 95.2 | 92.2 | 94.8 | 93.5 |
| SPEL-base (no mention-specific c. set) | 94.6 | 94.4 | 94.5 | 92.5 | 90.1 | 91.2 |
| SPEL-base (using PPRforNED c. set - context-agnostic) | 98.3 | 91.6 | 94.8 | 98.3 | 86.4 | 92.0 |
| SPEL-base (using PPRforNED c. set - context-aware) | 99.4 | 90.9 | 95.0 | 99.4 | 84.9 | 91.6 |

Table 2: Mention Detection evaluation results of SPEL in comparison to the work of De Cao et al. (2021a) using their released evaluation code (from utils.py). As De Cao et al. (2021a) use PPRforNED candidate sets, we only compare the SPEL results calculated using these candidate sets.

| Approach | EL Micro-F1 | | US$ for |
| --- | --- | --- | --- |
| | testa | testb | 1000 docs |
| GPT-3.5 (zero-shot) | 47.3 | 52.9 | 4.22 |
| GPT-4.0 | | | |
| (zero-shot) | 40.4 | 54.1 | 42.17 |
| (few-shot w/ CoT) | 62.4 | 66.2 | 59.37 |
| SPEL-base | 91.3 | 85.5 | 2.28 |
| SPEL-large | 91.6 | 85.8 | 2.64 |

Table 3: Comparison of the performance of SPEL (in no *mention-specific* candidate set setting) to zero and few shot GPT-3.5-turbo-16k (accessed on June 16, 2023) and GPT-4-0613 (for the best performing prompts we attempted; accessed on August 24, 2023). For few-shot experiments we constructed the prompt using the chain-of-thought (CoT) method of Wei et al. (2022).

| | Approach | EL Micro-F1 | | |
| --- | --- | --- | --- | --- |
| | | testa | testb | testc |
| SPEL-base | no mention-specific c. set | 89.6 | 82.3 | 73.7 |
| | KB+Yago c. set | 89.5 | 83.2 | 57.2 |
| | PPRforNED c. set | | | |
| | context-agnostic | 90.8 | 84.7 | 45.9 |
| | context-aware | 91.8 | 86.1 | - |
| SPEL-large | no mention-specific c. set | 89.7 | 82.2 | 77.5 |
| | KB+Yago c. set | 89.8 | 82.8 | 59.4 |
| | PPRforNED c. set | | | |
| | context-agnostic | 91.5 | 85.2 | 46.9 |
| | context-aware | 92.0 | 86.3 | - |

Table 4: Entity Linking evaluation results of SPEL with a *fixed candidate set* size of 500K over AIDA test sets. Since the *context-aware* candidate sets require a mechanism for generating/looking up the candidate set during inference, we do not evaluate testc in this setting.

## 5 Conclusion

We introduced several improvements to a structured prediction approach for entity linking. Our experimental results on the AIDA dataset show that our proposed improvements to the structured prediction model for entity linking can achieve state-of-the-art results using a commonly used evaluation toolkit providing head to head numbers for competing methods on the same dataset. We show that our approach has the best F1-score on this task compared to the state-of-the-art on this dataset. We are more compute efficient with many fewer parameters and our approach is also much faster at inference time, providing faster throughput, compared to previous methods.

In future work, we plan to extend our work to other entity linking datasets in other domains such as biomedical research. We also plan to research multilingual applications of structured prediction for entity linking including benefits to projection of entity linking concepts from one language to another and using multilingual representation learning for our base model.

dataset, for over two decades we acknowledge the potential concern of adaptive overfitting. In response to this concern, we used their newly annotated NER test set of 131 Reuters news articles published between December 5th and 7th, 2020. We meticulously linked the named entity mentions in this test set to their corresponding Wikipedia pages, using the same linking procedure employed in the original AIDA dataset. Our new entity linking test set, AIDA/testc, has 1,160 unique Wikipedia identifiers, spanning over 3,777 mentions and encompassing a total of 46,456 words.

We re-evaluate SPEL across all four settings outlined in Section 4.4 using the 500K entity output vocabulary and over all three AIDA test sets: testa, testb, and testc. We report our findings in Table 4. Examining the results shows that our newly created testc presents a new challenge for entity linking because the currently available candidate sets prove unhelpful and, in fact, detrimental to entity linking. The SPEL-large results for testa and testb show that SPEL with an unconstrained *fixed candidate set* size still matches the performance of the best model published before SPEL (with *fixed candidate set*s).

## Limitations and Ethical Considerations

Like other deep learning-based entity linking systems, SPEL relies on a predefined *fixed candidate set*. This implies that the user needs prior knowledge about the entities which they seek (e.g., `Barack_Obama`) and must include those entities in the model's output mention vocabulary. It is important to note that SPEL is unable to detect entities that are not included in its defined *fixed candidate set*. Expanding SPEL to support zero-shot entity linking is an area that we leave for future exploration and development. Our research is on English only, and we acknowledge that entity linking for other languages is also relevant and important. We hope to extend our work to cover multiple languages in the future. We inherit the biases that exist in our training data and we do not explicitly de-bias the data. It is possible that certain types of entities are under-represented in this dataset, so care must be taken before using our model for general purpose entity-linking that it is re-trained on a suitably de-biased training dataset that compensates for the fact that some under represented entities might be infrequent or missing from our training data. We are providing our models and code to the research community and we trust that those who use the model will do so ethically and responsibly.

## Acknowledgements

We would like to thank the anonymous reviewers for their helpful comments. The research was partially supported by the Natural Sciences and Engineering Research Council of Canada grants NSERC RGPIN-2018-06437 and RGPAS-2018-522574.

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

## A Wikipedia Redirect Normalization

van Hulst et al. (2020) report better results using an older Wikipedia dump from 2014 compared to the dump from 2019. One possible explanation for this finding is that the 2014 dump contains Wikipedia entries with page identifiers that are more closely aligned with the annotated data. Over time, page identifiers in Wikipedia have undergone changes, and some of the older identifiers used in annotating test datasets may now function as redirect links. To tackle this issue, researchers such as Broscheit (2019) and Yamada et al. (2020b, 2022) have considered redirect link normalization. We follow the same approach and use the collection of Wikipedia redirect links[13] to find all the redirect pairs $(u, v)$ where $u$ is not in our *fixed candidate set* and $v$ is in the set. In inference, whenever SPEL predicts $u$, we automatically replace it with $v$.

## B More on Comparison to OpenAI GPT Models

In this section, we provide more information on our experimental procedure which can help replicating our results.

When utilizing a pre-trained GPT language model, it is common practice to structure the task description as a prompt, which is then passed to the model. The model leverages its linguistic understanding to generate a solution in the form of a completion, based on the given prompt. However, a crucial limitation arises in the process of identifying the appropriate prompt, as its selection greatly influences the successful completion of the task.

### B.1 Zero-shot experiments

Our best performing prompt was as follows:

```
You are a Wikipedia annotator.
Annotate the Wikipedia entities
in the following paragraph, and
produce the output in markup
using the  element and the
data-entity attribute:
```

In each query[14], in the line after the prompt, we add the AIDA document received from GERBIL and pass it to GPT using `openai.ChatCompletion.create` API[15]. In cases where the documents exceed the maximum subword limit of GPT, we employ `spacy` python library, to divide the document, and create

---

[13] http://downloads.dbpedia.org/2016-10/core-i18n/en/redirects_en.ttl.bz2

[14] Cho et al. (2022) employ GPT for entity linking by implementing a process that involves a sequence of summarization and multiple-choice queries to GPT. However, we have found this approach to be rather costly. Furthermore, it necessitates prior knowledge of the target mention to condition the summary accordingly. Additionally, it relies on a set of candidates generated through heuristics which undermines the feasibility of utilizing GPT for end-to-end entity linking.

[15] We used python's `openai` package version "0.27.6" and we set `temperature=0`, `top_p=1`, `frequency_p enalty=0.`, `presence_penalty=0`.

query prompts consisting of approximately 1000 tokens each. Subsequently, we concatenate the received responses to these queries to form a comprehensive result. Following this, we parse the generated markup and associate each annotation with its corresponding segment in the original text. We proceed by forwarding the predicted annotations that match entries in the Wikipedia *fixed candidate set*, along with the extracted spans, back to GERBIL.

During the experiments, we analyzed the validation set results and observed consistent patterns that shed light on the challenges posed by generative language models in entity linking. One notable observation was the presence of annotations from a mixture of knowledge bases and domains, indicating that the model possesses an excessive amount of knowledge, leading to *distractions* in the annotation process focused on entity linking over Wikipedia. With this regards, we observed a lack of stability in the model's output even when setting the `temperature` parameter to 0. Despite using the same prompt, the model occasionally confused entity linking (EL) with named entity recognition (NER) and reported mentions annotated with NER tags such as `Person` or `Location`. In our experiments, we removed all predicted spans with such tags and did not relay them back to GERBIL.

Furthermore, due to the nature of generative models, there were instances where the model failed to generate the complete entity, resulting in incomplete predictions (for example it generated `Leicestershire` instead of the full entity identifier `Leicestershire County Cricket Club` or `Pohang` instead of `Pohang Steelers`). In these instances, if an exact match to an entity in the knowledge base was not found, we collected all entities in the *fixed candidate set* that included the full prediction from the generative language model. From this collection[16], we randomly selected one of those mentions and reported it back to GERBIL instead of the original prediction generated by the model.

### B.2 Few-shot experiments

In light of the growing popularity of few-shot prompting techniques with GPT language models, we conducted a survey of some of the leading approaches for experimenting with few-shot ex-

---

[16]In the majority of cases, the candidate set contains only one element if it is not empty.

amples. Given the promising outcomes associated with the chain-of-thought (CoT; Wei et al., 2022) prompting technique, we chose to conduct our few-shot experiments using this particular method. To construct our best performing few-shot CoT prompt, we retained our zero-shot prompt and added the following lines as an extension:

```
Document: "EU rejects German
call to boycott British lamb."
Answer: <p> <chain-of-thought>
Considering EU, German, and
British are shown in the text
together with the word boycott,
this is a political document,
I should annotate EU with
"European Union", German with
the country "Germany", and
British with the country "United
Kingdom". I make sure I do not
mistake Wikipedia identifiers
with entity type identifiers,
for example I choose "United
Kingdom" instead of the incorrect
general entity type "country".
I make sure to annotate all
entities even if there is
a large number of entities.
</chain-of-thought> <result>
 EU  rejects  German
 call to boycott 
British lamb.</result></p>
```
Adding more examples in the same format did not significantly improve performance, but it substantially increased the prompting cost to GPT-4.

We maintained the rest of the configurations and setups for the few-shot experiments the same as we had in the zero-shot experiments.

## C Out-of-domain Experiments and Results

Few of the publications listed in Table 1 recommend assessing entity linking models on *out-of-domain* testing datasets. These datasets typically lack associated training sets and are often annotated with entity links to variations or subsets of the `DBpedia` (Auer et al., 2007) knowledge base. Out-of-domain annotation typically operates under the assumption that the knowledge base entry iden-

| Approach | MSNBC | Derczynski | KORE | $N^3$ Reuters | $N^3$ RSS | OKE2015 | OKE2016 |
|---|---|---|---|---|---|---|---|
| Hoffart et al. (2011)† | 65.1 | 32.6 | 55.4 | 46.4 | 42.4 | 63.1 | 0.0 |
| Kolitsas et al. (2018) | 72.4 | 34.1 | 35.2 | 50.3 | 38.2 | 61.9 | 52.7 |
| van Hulst et al. (2020) | **74.4** | 41.2 | 61.6 | 49.7 | 34.3 | **64.8** | **58.8** |
| De Cao et al. (2021b) | 73.7 | 54.1 | 60.7 | 46.7 | 40.3 | 56.1 | 50.0 |
| Zhang et al. (2022) | 72.1 | 52.9 | **64.5** | **54.1** | 41.9 | 61.1 | 51.3 |
| SPEL-base | 64.5 | 50.7 | 48.7 | 47.9 | 41.9 | 55.9 | 57.4 |
| SPEL-large | 63.1 | **59.1** | 53.7 | 47.1 | **44.4** | 59.5 | 56.6 |
| Oracle‡ | 93.2 | 91.4 | 99.6 | 99.7 | 98.0 | 88.2 | 91.4 |

Table 5: Comparison of SPEL (with a *fixed candidate set* size of 500k) evaluation results with the literature on out-of-domain datasets. The best score is shown as **bold** and the second best is shown as underlined.
†Results from (Kolitsas et al., 2018 - Table 2).
‡The "Oracle" results are calculated through feeding the gold annotations of each dataset to GERBIL, and depicts the In-KB annotation quality of each dataset.

tifiers remain consistent between in-domain and out-of-domain scenarios. While this assumption may hold true to a certain extent, as DBpedia's primary focus has been on information extraction from Wikipedia, it's important to note that the temporal evolution of both knowledge bases has introduced discrepancies. These datasets, which are between 8 to 16 years old at the time of writing this paper, have been affected by temporal changes, and the two knowledge bases are not always perfectly aligned. The following offers a concise overview of some of the most commonly utilized out-of-domain datasets for evaluation:

**MSNBC** (Cucerzan, 2007) contains 20 MSNBC news stories (annotated with Wikipedia) from different categories including Health, Technology, Sports, etc.

**KORE** (Hoffart et al., 2012) contains 50 sentences annotated with DBpedia and chosen from five domains: celebrities, music, business, sports, and politics. It was created to examine the disambiguation functionality in the older entity disambiguation models.

**$N^3$ Reuters** and **$N^3$ RSS** (Röder et al., 2014) contain mentions referring to persons, places and organizations (DBpedia annotations). The Reuters dataset contains 128 news stories from Reuters news agency and the RSS dataset contains 500 RSS feed messages from worldwide newspapers (in English).

**Derczynski** (Derczynski et al., 2015) contains 182 tweets annotated with DBpedia knowledge base entities.

**OKE challenge 2015 and 2016** evaluation sets (Nuzzolese et al., 2015) contain 101 and 55 sentences from Wikipedia articles (reporting biographies of scholars), respectively, annotated using a mixture of annotations from DBpedia and the OKE entity identifiers.

We provided the *out-of-domain* data sets to SPEL, using a *fixed candidate set* of 500K entities, and compared its performance against other methods that have reported results on these datasets. The comparative results can be found in Table 5.

Please note that SPEL's tokenization procedure does not allow the generation of annotations that start or end within a single word (separated by space characters). For instance, in SPEL, the token `washington-based` is considered a single word, whereas out-of-domain datasets contain several annotations that commence or conclude within a single word. Additionally, each dataset necessitates a specific redirect normalization schema; for example, `China` is annotated as `People's_Republic_of_China` in KORE, but in $N^3$ RSS, it is annotated as `China`.

Nevertheless, SPEL-large delivers the best results on two out of seven and the second-best result on one out of seven test sets. It doesn't significantly underperform the other models in terms of performance on the remaining four test sets.