# OpenReview forum: "SpEL: Structured Prediction for Entity Linking"
_EMNLP/2023/Conference — EMNLP 2023 Main_

### Official Review · Reviewer_FVrb · 2023-07-30

**Soundness:** 4

**Excitement:**

4: Strong: This paper deepens the understanding of some phenomenon or lowers the barriers to an existing research direction.

**Paper Topic And Main Contributions:**

This paper proposes a structured prediction approach to entity linking where the goal is to link appropriate text to the entitles from the surface forms that represent those entities. The paper claims several contributions including a new structured prediction framework, a context sensitive prediction aggregation strategy, in-domain mention vocabulary, a model fine-tuning on the tokenized sequence to make it more capable of recognizing mentions, improved model parameter and speed 	efficiency, and an improvement to the GERBIL platform to run reproducible experiments.

**Questions For The Authors:**

A) Could you please clarify whether the context sensitive prediction aggregation strategy can be utilized in a setting where the right answer is not added to the candidate set?

**Reasons To Accept:**

Entity linking is an important and challenging problem, and the approach proposed in this paper  performs better then prior approaches. In addition, while the community tends to evaluate in the unnatural setting of ensuring that the right answer is among the candidate set, the proposed approach also performs quite well when no candidate set is used. In addition the improvement to the GERBIL platform to make it easier to integrate with a python based approach is an important contribution of the paper. Finally, the paper makes a nice discovery that fine-tuning on a large entity-linking dataset which is re-tokenized without the knowledge of mention spans after fine-tuning on the dataset with linking to the knowledge base improves performance because the final fine-tuning step matches the way the model will encounter text when it is performing the task.

**Reasons To Reject:**

One of the main contributions of this is a context sensitive prediction aggregation strategy; however, it only appears to be used in the setting where the candidate set always includes the correct answer. While this artificial setting is commonly used in the literature, it is important to also report results when the right answer is only present because the model included it the candidate set. This would be help determine the importance of this contribution. As the paper is currently written one cannot determine if this is a useful technique for a real-world setting.

Another contribution is the in-domain fixed vocabulary, which reduces the model size to prevent the model from generating entities that are not part of the in-domain data. This also seems problematic. While this scenario clearly occurs in the AIDA dataset where only a fraction of Wikipedia entries occur the text, having such knowledge in a real world setting needs to motivated. Otherwise, you are just making the problem easier in the artificial environment of the evaluation.

A final weakness concerns reproducibility. It would be very difficult for another researcher to reproduce this work from the description in the paper. Unless it was missed, no mention was made to releasing the code and there was no supplementary data indicating the intent to release the code.

The authors can address these concerns using the explanations provided in the response to reviewers.

**Reproducibility:**

5: Could easily reproduce the results.

**Reviewer Confidence:**

2: Willing to defend my evaluation, but it is fairly likely that I missed some details, didn't understand some central points, or can't be sure about the novelty of the work.

**Typos Grammar Style And Presentation Improvements:**

It appears that some of the approaches (yours and others) use a candidate set and others do not. In addition, it appears that many only report results on the augmented fixed candidate set that always includes the right answer. It would be useful if Table 1 would identify which techniques use an augmented fixed candidate set so help understand which comparisons are fair comparisons.

Figure 1 would be greatly improved by using real words and entities rather than word1..word5, and c1, c2, etc. In addition saying that c1 is not part of the candidate set rather than that c2 is the only one in the candidate set would make it easier to follow the example. In particular, a candidate set of size 1 is problematic because there is no point to generate a ranking for one candidate. It should just be the answer.

L176 ; however,
L331 remove the
L390-394 Awkward: rewrite. A sentence in the form of “Because X, our methods does not need to A or B” would be more clear. Be sure that X is more specific than “this strategy”. What part of the strategy means that A and B are unnecessary.
L632 much less -> many fewer
L633 must -> much

---

> ### Author Rebuttal · Authors · 2023-08-27
>
> > One of the main contributions of this is a context sensitive prediction aggregation strategy; however, it only appears to be used in the setting where the candidate set always includes the correct answer. While this artificial setting is commonly used in the literature, it is important to also report results when the right answer is only present because the model included it the candidate set. This would be help determine the importance of this contribution. As the paper is currently written one cannot determine if this is a useful technique for a real-world setting. Could you please clarify whether the context sensitive prediction aggregation strategy can be utilized in a setting where the right answer is not added to the candidate set?
>
> We acknowledge the reviewer's thoughtful observation regarding the real-world applicability of our context-sensitive prediction aggregation strategy. Ensuring the relevance of our contributions to practical scenarios has been a paramount concern for us throughout this research. Our paper addresses this concern by conducting experiments in two distinct settings: one with semi-automatically curated candidate sets and the other without them. In the latter setting, the softmax in the final prediction layer is over a set of entities which we refer to as the "fixed candidate set". Furthermore, we also use a compute-expensive fine-tuning step in Section 3.3 that is done on the entire Wikipedia, not just on the AIDA dataset to allow our model to be trained and tested on different Wikipedia-based EL datasets with and without candidate sets.
>
> Our experimental settings (both with and without hand-selected candidate sets) do not presume the inclusion of the correct answer in the candidate set (if used). In fact, as we explain in lines 558-570 of the submitted paper, using candidate sets can sometimes hinder performance because they may not encompass all the correct answers. As an example, consider the KB+Yago dataset, generated by examining the co-occurrence of mentions and Wikipedia links across the entire Wikipedia. In a real-world scenario, when linking "american composer" in a news story about "John Luther Adams," who is another American composer, using a candidate set like [["John_Adams_(composer)", 0.875], ["United_States", 0.125]] leads to error. Our "SpEL (no mention-specific candidate set)" results do not make any assumptions about availability of candidate sets and choose from the entirety of the in-domain entity identifiers, which is certainly a realistic setting for real-world applications.
>
> We can obtain higher results sometimes without a candidate set, as we can see from the results using KB+Yago candidate set on AIDA test-a in Table 1 and also the RoBERTa-large results in the same setting in our response to reviewer `8Rzo`.
>
> We include the results using candidate sets mainly because previous papers have reported evaluation scores only using candidate sets (different candidate sets in different papers). We report our results with and without candidate sets for a fair comparison to previous work.
>
> > Another contribution is the in-domain fixed vocabulary, which reduces the model size to prevent the model from generating entities that are not part of the in-domain data. This also seems problematic. While this scenario clearly occurs in the AIDA dataset where only a fraction of Wikipedia entries occur the text, having such knowledge in a real world setting needs to motivated. Otherwise, you are just making the problem easier in the artificial environment of the evaluation.
>
> We recognize the reviewer's concern regarding the in-domain fixed vocabulary and its relevance to real-world settings. We'd like to clarify that the concept of a fixed candidate set is not inherently tied to specific datasets or data dependencies. While it can be derived from entities linked in the in-domain data, it can also be constructed from any subset of knowledge base entities that are pertinent to the task at hand. Furthermore, all previous research into entity linking makes the same assumptions that we do. In addition, unlike many previous papers, we also report our results without a constrained candidate set.
>
> Therefore, we include experiments with hand-crafted candidate sets as a way to have a fair comparison with prior published work in this area. However, we think research groups (e.g. from Facebook) who have published in this area do have some real-world examples in mind. To illustrate this with a simplified real-world example, imagine a company's interest lies solely in recognizing the names of football teams. In this scenario, the company can easily compile an exhaustive list of all football teams from Wikipedia (e.g., https://en.wikipedia.org/wiki/Coastal_Carolina_Chanticleers_football, https://en.wikipedia.org/wiki/Kansas_State_Wildcats_football, and so on) and employ this set as their fixed candidate set within our SpEL framework. The use of the in-domain set of mentions in our experiments was intended to represent this scenario, not as a rigid requirement to generate the fixed candidate set exclusively from in-domain data. The flexibility to create such sets tailored to specific real-world applications is a key feature of our approach. Our work, and prior related work in this area extends beyond artificially simplifying the problem; it offers adaptability to diverse real-world contexts where different subsets of entities may be of interest.
>
> We also emphasize that we also provide results without candidate sets which to the best of our knowledge are better than prior results without candidate sets that use the same dataset (compare with De Cao et al. 2021a without candidate sets in Table 1). SoTA results like EntQA (Zhang et al, 2022) and (Feng et al, 2022) use methods to compute/use a candidate set. EntQA does a document level search for candidate entities, and (Feng et al, 2022) says that they "use KnowBert’s candidate generator to first find all spans that might be potential entities in a sentence. These spans are matched in a precomputed spanentity co-occurrence table (Hoffart et al., 2011) and each span is annotated with linked entity candidate IDs associated with prior probabilities based on frequency." The main result in our paper is that structured prediction is a more effective and less compute-intensive method to get SoTA results for entity linking. Our method is a lot simpler and easier to implement from scratch than many of the existing approaches because we can effectively exploit the bidirectional context more efficiently than prior work.
>
> > A final weakness concerns reproducibility. It would be very difficult for another researcher to reproduce this work from the description in the paper. Unless it was missed, no mention was made to releasing the code and there was no supplementary data indicating the intent to release the code.
>
> While we appreciate the reviewer's concern, we already state in the footnote on page 2 that, when accepted, we will release the training data, pre-trained models and our source code for training and inference (including the settings with and without candidate sets). In addition, we also plan to release our new test set for entity linking as explained in further detail in our response to reviewer `8Rzo`.
>
> Thank you for you feedback on grammar, style, and presentation. If accepted, we are committed to implementing the recommended improvements in the camera-ready version.

---

### Official Review · Reviewer_8QvA · 2023-08-02

**Soundness:** 5

**Excitement:**

5: Transformative: This paper is likely to change its subfield or computational linguistics broadly. It should be considered for a best paper award. This paper changes the current understanding of some phenomenon, shows a widely held practice to be erroneous in someway, enables a promising direction of research for a (broad or narrow) topic, or creates an exciting new technique.

**Missing References:**

Could not identify missing references, but I am working on this particular topic (structured prediction) at the moment.

**Paper Topic And Main Contributions:**

The paper is generally well-written and well illustrated. The authors tackle the issue of structured prediction in the context of named entity linking.

The introduction does not clearly contain a plain English explanation of what is structured prediction. There is a definition in Section 3.2, but that one is more formal. This makes it less accessible to casual readers.

Overall, the formalization and definitions of NEL, spans, candidate sets (3.1), structured prediction (3.2), and fine-tuning procedure (3.3) are good and easy to understand.

The experimental section is rather good. The experiments are conducted with GERBIL and details about how to add new annotators (e.g., via a Java middleware called SpotWrapNifWS4Test). A minor issue from my point of view is represented by the comparison with GPT-3.5-turbo-16k, mainly due to the framing used (a setup from a 2021 work). Currently SoA has advanced significantly when it comes to few-shot learning for LLMs - and the paper does not contain any references to Chain of Thought, Tree of Thought and other similar problem-solving techniques. Granted, their technique is better than the straightforward output of GPT-3.5 in this particular instance, but we will not know for sure if that's really the case without testing at least several few-shot problem-solving techniques with GPT-3.5. The prompt used for this task, on the other hand is rather good.

The limitations section acknowledges some problems with zero-shot entity linking and multilingualism, but otherwise the method is generally sound and can be applied in various settings.

I have read the rebuttal and the authors have done an impressive job of addressing most comments.

**Questions For The Authors:**

Q1: Why were other LLMs left out of the evaluation? GPT-3.5 may be the best, but Claude 2 sometimes outperforms it.

**Reasons To Accept:**

- good structured prediction technique
- implemented with GERBIL, therefore reproducibility is not an issue
- results are better than expected

**Reasons To Reject:**

- the comparison with LLMs is rather thin - I see no reason to include it, but if it's included it needs to be done properly
- too focused on the NER/NEL problems and no consideration is given to the larger issue of slot filling

**Reproducibility:**

5: Could easily reproduce the results.

**Reviewer Confidence:**

5: Positive that my evaluation is correct. I read the paper very carefully and I am very familiar with related work.

**Typos Grammar Style And Presentation Improvements:**

Overall the paper seems to be well-written and couldn't really identify major typos or grammar issues.

---

> ### Author Rebuttal · Authors · 2023-08-26
>
> > The introduction does not clearly contain a plain English explanation of what is structured prediction...
>
> Thank you for highlighting this. We will add additional context to the introduction to improve the paper's accessibility to a wider audience.
>
> > A minor issue from my point of view is represented by the comparison with GPT-3.5-turbo-16k, mainly due to the framing used (a setup from a 2021 work). Currently SoA has advanced significantly when it comes to few-shot learning for LLMs - and the paper does not contain any references to Chain of Thought, Tree of Thought and other similar problem-solving techniques...., the comparison with LLMs is rather thin - I see no reason to include it, but if it's included it needs to be done properly, Why were other LLMs left out of the evaluation? GPT-3.5 may be the best, but Claude 2 sometimes outperforms it.
>
> The field is moving very fast and at the time of submission, GPT-3.5 was the best GPT language model that was available to us and prompt engineering was still quite exploratory. We did try many different prompts since then with no improvements. Since our submission, OpenAI opened up GPT-4 for API access and there are new heuristics in writing prompts and as a result we were able to extend our experiments. The table below reports our results using GPT-4 and better prompting.
>
> | Approach                                         | EL Micro-F1 tetsa | EL Micro-F1 testb | US$ for 1K documents |
> |--------------------------------------------------|-------------------|-------------------|----------------------|
> | GPT3.5 (zero-shot)                               | 47.3              | 52.9              | 4.22                 |
> | GPT4.0 (zero-shot)                               | 40.4              | 54.1              | 42.17                |
> | GPT4.0 (few-shot with chain-of-thought prompts)  | 62.4              | 66.2              | 59.37                |
> | SpEL-base (no c. set)                            | 90.9              | 84.2              | 2.28                 |
> | SpEL-large (no c. set)                           | 91.4              | 85.8              | 2.64                 |
>
> Additionally, we thank the reviewer for bringing Claude 2 to our attention. We have initiated the process of obtaining API access from Anthropic to incorporate comparison results between Claude 2 and GPT-4 into our evaluation. We plan to add Claude-2 results in the camera-ready version, if accepted, provided that we receive the necessary API access from Anthropic.
>
> It is thought-provoking that despite any published results on improved entity linking using the instruction-tuned few-shot setting of large language models like GPT4 we had to pay to compare against such models and provide a comparison to the SoTA. This is a new source of expense for academic groups conducting research into undeniably useful NLP applications like entity linking that do not nicely fit into the chat and question-answering framework of instruction-tuned few-shot LLMs. We noticed that even GPT4 will give up on longer outputs and had to be coerced to produce output on the entire input sentence.
>
> > The paper is too focused on the NER/NEL problems and no consideration is given to the larger issue of slot filling.
>
> We appreciate the reviewer's feedback and their observation about the focus of our paper. While it is true that both Entity Linking (EL) and Slot Filling (SF) are part of the broader domain of information extraction from unstructured text, however, we see EL as a method of __alignment__ of mentions to an existing ontology such as Wikipedia. This can be helpful in addressing issues with hallucinations and truthfulness because facts about mentions in text can be linked to an ontology such as Wikipedia and the facts can be compared against a curated human expert on the entity/topic. In SF the objective is to extract specific information about an entity to populate predefined slots (as illustrated in Figure 1 of the KILT paper, https://aclanthology.org/2021.naacl-main.200, for example). It would be interesting to explore how EL can be combined with SF for a more comprehensive look at information extraction but unfortunately that was out of scope for our current paper.

---

### Official Review · Reviewer_8Rzo · 2023-08-07

**Soundness:** 3

**Excitement:**

3: Ambivalent: It has merits (e.g., it reports state-of-the-art results, the idea is nice), but there are key weaknesses (e.g., it describes incremental work), and it can significantly benefit from another round of revision. However, I won't object to accepting it if my co-reviewers champion it.

**Paper Topic And Main Contributions:**

This paper proposes a new entity linking system called SPEL. The new system employs three new ideas to improve the accuracy and efficiency of entity linking. Experiments on AIDA dataset is conducted, and SPEL is compared with several baselines. The results show that SPEL outperforms the compared approaches.

**Reasons To Accept:**

1.A new entity linking system is proposed, which outperforms compared approaches on AIDA dataset.

2.Several new methods are designed to enhance the performance of entity linking, including a context sensitive predication aggregation strategy, in-domain mention vocabulary for creating fixed candidate set, a fine-tune method without using explicit mention location information.


**Reasons To Reject:**

Experiment part needs to be enhanced. First, only one dataset is used in the experiments; more datasets should be tested. Second, there is no analysis on the influence of underlying language model to the entity linking results; for example, what are the results of SPEL if RoBERTa is replaced with BERT?

**Reproducibility:**

3: Could reproduce the results with some difficulty. The settings of parameters are underspecified or subjectively determined; the training/evaluation data are not widely available.

**Reviewer Confidence:**

3: Pretty sure, but there's a chance I missed something. Although I have a good feel for this area in general, I did not carefully check the paper's details, e.g., the math, experimental design, or novelty.

---

> ### Author Rebuttal · Authors · 2023-08-26
>
> > Experiment part needs to be enhanced. First, only one dataset is used in the experiments; more datasets should be tested.
>
>
> While setting up the experiments for this paper, we also struggled with finding more than the AIDA-CoNLL dataset for Wikipedia entity linking. Unfortunately, the other datasets we could find for Wikipedia entity linking were too small or had problematic out-dated annotations w.r.t. Wikipedia entities. In response to this situation, we have manually annotated a new dataset comprising 131 Reuters news articles basing on the NER dataset of (Liu and Ritter 2023; https://aclanthology.org/2023.acl-long.459) which was published after our submission. Our new dataset contains 1,145 unique new entity identifiers and spans over 4,028 mentions, encompassing a total of 46,456 words. We are in the process of the final annotation review and collecting results on current SoTA methods including ours. We plan to incorporate the evaluation results from this new dataset, along with the currently reported test sets from AIDA, if we are accpted in the camera-ready version of the paper. We believe this addition will enhance the comprehensiveness of our evaluation and address the reviewer's concern adequately.
>
>
> >Second, there is no analysis on the influence of underlying language model to the entity linking results; for example, what are the results of SPEL if RoBERTa is replaced with BERT?
>
> While we acknowledge the importance of exploring different underlying language models, our resources have been constrained by budget limitations (see Section 3.3 for our fine-tuning procedure). Despite this, we did conduct a secondary set of experiments with RoBERTa-large as the underlying model to study the effect of base model parameter size on the results.
>
>
> | Approach                                                                                        | EL Micro-F1 test-a          | EL Micro-F1 test-b  |           #params on GPU         | speed sec/doc  |
> |---------------------------------------------------------------------------------------|:--------------------------------:|:-------------------------:|:--------------------------------------:|:---------------------------:|
> | **SpEL-large** (no mention-specific candidate set)                       |          91.4                       |            85.8             |                 361.1M                  |       0.112        |
> | **SpEL-large** (KB+Yago candidate set)                                       |          90.6                       |            85.8             |                 361.1M                  |       0.157        |
> | **SpEL-large** (PPRforNED candidate set) (context-agnostic)     |          92.5                       |            87.4             |                 361.1M                  |       0.156        |
> | **SpEL-large** (PPRforNED candidate set) (context-aware)        |          **93.2**                        |            **88.4**             |                 361.1M                  |       0.156        |
>
>
> Please note that the SpEL-large parameter size on GPU is due to the size of the underlying RoBERTa-large model (355.4M parameters) and the classification layer (5.7M parameters). The model speed remains in the range of of 0.1 to 0.16 seconds on average to process a news document even for the larger model. The new results push our previously reported state-of-the-art results by 0.8 and 0.9 Micro-F1 scores higher for AIDA testa and testb respectively.

---

### Meta-Review · Area_Chair_wcPK · 2023-09-15

**Recommendation:** 4

**Metareview:**

The paper proposes to solve entity linking from a new perspective of structure prediction. To to this, the authors design a fixed vocabulary strategy to reduce the model prediction space, and address the training/inference mismatch issue by 2-stage finetuning. After serious discussion and consideration, the reviewers' main concerns are about the datasets (GERBIL is a platform easily scalable to new datasets), baselines (comparison with LLMs), as well as the above two contributions. The authors actively provide more evidence which we think has mostly solved the concerns.

Overall, we appreciate both the efforts of reviewers and authors. This work is ready to publish. We hope the authors can continue to improve the paper according to the comments and your response.

---

### Decision · Program_Chairs · 2023-10-07

**Decision:**

Accept-Main

**Comment:**

The paper proposes to solve entity linking from a new perspective of structure prediction. To to this, the authors design a fixed vocabulary strategy to reduce the model prediction space, and address the training/inference mismatch issue by 2-stage finetuning. After serious discussion and consideration, the reviewers' main concerns are about the datasets (GERBIL is a platform easily scalable to new datasets), baselines (comparison with LLMs), as well as the above two contributions. The authors actively provide more evidence which we think has mostly solved the concerns.

Overall, we appreciate both the efforts of reviewers and authors. This work is ready to publish. We hope the authors can continue to improve the paper according to the comments and your response.